**Citation:** Østergaard FG, Penninx BWJH, Das N, Arango C, van der Wee N, Winter-van Rossum I, et al. (2024) The aperiodic exponent of neural activity varies with vigilance state in mice and men. PLoS ONE 19(8): e0301406. https://doi.org/10.1371/journal.pone.0301406

**Data Availability Statement:** The mouse data can be found on GIN, with the following DOI: 10.12751/g-node.j2h8vr For access to the human data, a data access request form has to be sent to Martien Kas

# The aperiodic exponent of neural activity varies with vigilance state in mice and men

Freja Gam Østergaard[ID][1]*, Brenda W. J. H. Penninx[2], Neetha Das[3], Celso Arango[4,5], Nic van der Wee[6,7], Inge Winter-van Rossum[8], Jose Luis Ayuso-Mateos[9,10,11], Gerard R. Dawson[12], Hugh Marston[13,14], Martien J. H. Kas[ID][1]*

1 University of Groningen, Groningen Institute for Evolutionary Life Sciences, University of Groningen, Groningen, The Netherlands, 2 Department of Psychiatry and Amsterdam Neuroscience, VU University Medical Center, Amsterdam, The Netherlands, 3 Biotrial, Rennes, France, 4 Child and Adolescent Department, Institute of Psychiatry and Mental Health, Hospital General Universitario Gregorio Marañón, Madrid, Spain, 5 CIBERSAM, IiSGM, School of Medicine, Universidad Complutense, Madrid, Spain, 6 Department of Psychiatry, Leiden University Medical Center, Leiden, The Netherlands, 7 Leiden Institute for Brain and Cognition/Psychiatric Neuroimaging, Leiden University Medical Center, Leiden, The Netherlands, 8 Department of Psychiatry, University Medical Center Utrecht Brain Center, Utrecht University, Utrecht, The Netherlands, 9 Department of Psychiatry, Centro de Investigación, Universidad Autónoma de Madrid, Madrid, Spain, 10 Biomédica en Red de Salud Mental, CIBERSAM, Instituto de Salud Carlos III, Madrid, Spain, 11 Hospital Universitario de La Princesa, Instituto de Investigación Sanitaria Princesa (IIS-Princesa), Madrid, Spain, 12 P1vital, Wallingford, Oxfordshire, United Kingdom, 13 Boehringer Ingelheim Pharma GmbH & Co KG, CNS Diseases Research, Biberach an der Riss, Germany, 14 External Neurodegenerative Research, Eli Lilly and Company, Windlesham, United Kingdom

* fgoestergaard@health.sdu.dk (FGØ); m.j.h.kas@rug.nl (MJHK)

## Abstract

Recently the 1/f signal of human electroencephalography has attracted attention, as it could potentially reveal a quantitative measure of neural excitation and inhibition in the brain, that may be relevant in a clinical setting. The purpose of this short article is to show that the 1/f signal depends on the vigilance state of the brain in both humans and mice. Therefore, proper labelling of the EEG signal is important as improper labelling may obscure disease-related changes in the 1/f signal. We demonstrate this by comparing EEG results from a longitudinal study in a genetic mouse model for synaptic dysfunction in schizophrenia and autism spectrum disorders to results from a large European cohort study with schizophrenia and mild Alzheimer's disease patients. The comparison shows when the 1/f is corrected for vigilance state there is a difference between groups, and this effect disappears when vigilance state is not corrected for. In conclusion, more attention should be paid to the vigilance state during analysis of EEG signals regardless of the species.

## Introduction

Pink noise or 1/f signals is a common phenomenon in signal processing and in electroencephalography (EEG) especially and may have a potential as a biomarker for psychiatric disorders [1]. The origin of 1/f signals is not fully understood, but it has been hypothesized as the typical response of any biosystem to white noise [2].

(corresponding author), who will facilitate communication to the PRISM coordination team in order to ensure compliance with the ethical conditions under which the data was collected.

**Funding:** 'This research was supported by the PRISM project (www.prism-project.eu) which has received funding from the Innovative Medicines Initiative 2 Joint Undertaking under grant agreement No 115916. This Joint Undertaking receives support from the European Union's Horizon 2020 research and innovation program and EFPIA. This publication reflects only the authors' views neither IMI JU nor EFPIA nor the European Commission are liable for any use that may be made of the information contained therein.'

**Competing interests:** Neetha Das, Hugh Marston and Gerard R. Dawson worked for a company during the data collection of the PRISM1 study. The companies provided support in the form of salaries for authors, but did not have any additional role in the study design, data collection and analysis, decision to publish, or preparation of the current manuscript which was performed as a secondary analysis using the PRISM1 data set. The specific roles of these authors are articulated in the 'author contributions' section.' Dr. Arango has been a consultant to or has received honoraria or grants from Acadia, Angelini, Biogen, Boehringer, Gedeon Richter, Janssen Cilag, Lundbeck, Medscape, Menarini, Minerva, Otsuka, Pfizer, Roche, Sage, Servier, Shire, Schering Plough, Sumitomo Dainippon Pharma, Sunovion, Takeda and Teva. The other authors have declared that no competing interests exist.

There are several proposed methods for extracting pink noise from an EEG signal [3, 4]. The present study is a response to the 2020 paper by Donoghue and colleagues, establishing the Fitting Oscillation and One Over F (FOOOF) pipeline and showing the 1/f as having a 'flatter' slope in older humans compared to younger [3]. This slope is also referred to as the aperiodic exponent. It has been suggested that this may be an expression of the balance between neuronal excitation and inhibition [5, 6]. The name 'aperiodic' suggests that it is unaffected by oscillatory activity and Donoghue's data analysis was performed without taking variation in vigilance state into account. In our study, we use FOOOF on scored local field potential data from mice and human, we show a significant effect of vigilance state on the aperiodic exponent.

The balance between excitation and inhibition (E/I balance) and especially imbalances have been linked to psychiatric illnesses like autism spectrum disorder [7–9] and schizophrenia [10] along with neurological disorders like Alzheimer's disease [5].

To illustrate the idea of vigilance state masking potential biomarkers for psychiatric and neurological diseases, the human data belongs to the transdiagnostic PRISM1 project containing data from people with either schizophrenia, probable Alzheimer's disease or age-matched controls. The mouse model used in the present study has a knockout of *Nrxn1α*. Loss-of-function mutations of the *NRXN1* gene are related to autism as well as schizophrenia [11]. Nrxn1 is a large presynaptic protein, and well-conserved across species [12]. As animals cannot be diagnosed *per se*, genotype serves as a surrogate for diagnosis. Only males are included in this manuscript, as male mice have previously showed a behavioral phenotype while female mice didn't [13]. Sex-differences are also found in psychiatric diagnosis in humans, including but not limited to E/I imbalance in autism [7].

## Methods

### Animal work

Twenty-two male mice of the transgenic Nrxn1α line [13], (genotypes: five wildtypes (wt), eight heterozygous knock-out (het) and nine homozygous knock-out (hom) mice) had electrodes implanted over the prefrontal cortex, along with bilaterally in the visual and auditory cortices. 24 hour recordings were carried out with a wireless transmitter in the home cage environment.

The animals were obtained from the *in-house* breeding facility. They were group-housed until surgery, thereafter housed in pair, with a separator to allow for smell and touch, but preventing damage to the implant. The cages contained woodchip bedding and enrichment in the form of a red, transparent hide-away and nesting material. Food and water were available *ad libitum*. The light/dark cycle was set to 12:12, lights out at 14:00.

All animal work was carried out in accordance with the EU directive 2010/63/EU. A prospective approval was given by the national authority (Centrale Commisie Dierproeven) license: AVD1050020198764 and the specific protocol was approved by the local body (Instantie van Dierenwelzijn) at Rijksuniversiteit Groningen. Surgery was performed under isoflurane anesthesia with carprofen analgesia, and all efforts were made to minimize suffering.

### EEG surgery

The animals underwent stereotactic surgery for electrode implantation at six weeks of age. Isoflurane anesthesia (induced at 5% and maintained at ~1.3%) was used during surgery. The head of the animal was shaved and wiped with 0.5% chlorhexidine (in 70% ethanol). The animal was fixed in a stereotaxic frame. Carprofen (5 mg/kg) was administered in the flank, for general analgesia and local analgesia lidocaine was injected SC on the head. An incision was

made and the scalp was cleared of tissue and remaining lidocaine. The skull was scratched with a scalpel and 35% phosphoric acid was applied. After 5 min the skull was cleaned and bilateral craniotomies were made over the visual cortex AP -4 ML ± 2.5, the prefrontal cortex AP 2.58 ML -1.57 and the auditory cortex AP -2.5 ML ± 3.5. The reference electrode was placed over cerebellum AP -6 ML 0. The electrodes were jeweller screws (0.7 mm, Antrin, USA), placed in each of the craniotomies and connected to the connector. Two stranded electrodes were inserted under the neck muscles to record muscular activity. The wires for the electrodes were preassembled into a plastic connector (ND associates, UK). The connector was attached to the scull using dental cement (3M, USA) the skin was closed around the cement with sutures. Analgesia in the form of carprofen (5mg/kg) was administered at 24 hours post-surgery. The animal had two weeks of recovery before any recordings were carried out.

## Recordings

Electrophysiological recordings were made over 24 consecutive hours when the mice were 8 and 22 weeks of age. Recordings were carried out with a Taini transmitter (Tainitec, UK) in the home cage environment. The data from the mice was sampled at 1084.7 Hz and online filtered (low-pass = 9700 Hz, high-pass = 0.35 Hz) using TainiLive. Subsequently it was exported as an.edf-file and scored according to vigilance state using custom scripts written in Matlab v2020a (Mathworks, US). The scripts are available on github: https://github.com/FrejaGam/EEGcode/wiki/Scoring-of-vigilance-states

The mouse data can be found on GIN, with the following DOI: 10.12751/g-node.j2h8vr.

For access to the human data, a data access request form has to be sent to Martien Kas (corresponding author), who will facilitate communication to the PRISM coordination team in order to ensure compliance with the ethical conditions under which the data was collected.

## Human data

Access to human data from the PRISM1 project was approved in July 2023. The data set contained 160 EEG recordings of men and women along with data on sex, age, diagnosis and site of recording.

Participants were recruited between July 2017 and March 2019 from five different recruitment sites across Spain and the Netherlands. The study was approved by the Ethics Review Board of all participating centers: Hospital General Universitario Gregorio Marañón, Hospital Universitario de La Princesa, University Medical Center Utrecht, VU University Medical Center Amsterdam and Leiden University Medical Center. All participants were given detailed information about the study before providing verbal and written consent. The rationale of the PRISM study is described in depth elsewhere: [14, 15].

Recordings were done using the 64-channel Waveguard™ system (ANT Neuro, Netherlands). The data was analyzed using the same pipeline as for the mice, however, more channels were selected and only the first 15 minutes of the recordings were used as this part contained the resting state paradigm. The age distribution of the PRISM1 data set is shown below, as a histogram of data points (electrodes of interest) (Fig 1).

## Data pipeline

All data was filtered using a butterworth filter of order 4 and high-pass: 0.5 Hz and low-pass: 70 Hz. Artifacts were rejected if they deviated more than 3 SD from the mean of the signal. The signal was scored into vigilance states on the basis of 1 s epochs. The mouse data was scored into seven vigilance states (three active states, REM and three nREM states), and the sleep category containing the nREM cycles in their natural sequence. The active states are

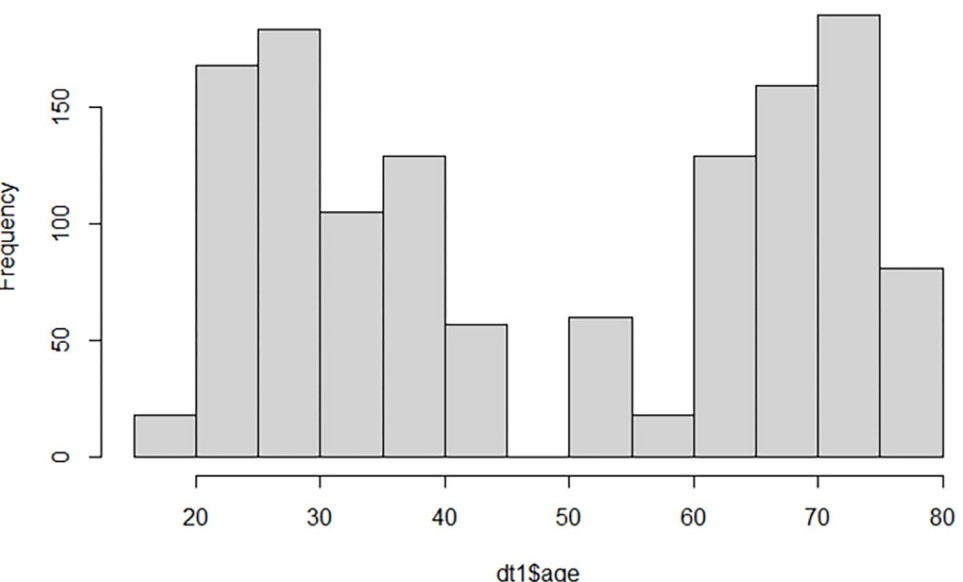

**Fig 1. Age distribution in the PRISM1 data set.** Age is shown in years, but binned per five years. Frequency (y-axis) accounts for the number of electrodes (12 per participant).

dominated by specific frequency bands. A1 is dominated by delta band activity [16] (0.2–3 Hz), a2 by theta band activity [17, 18] (3.2–6 Hz) and a3 by alpha band activity (6.2–12 Hz). A3 is more commonly known as resting state, while a1 is awake and active, and a2 has been associated with hippocampal activity [18]. The human data only included awake states and was scored into a1, a2 and a3.

After scoring the data the aperiodic exponent was computed using FOOOF described by Donoghue and colleagues. Power spectral density was computed in Matlab v2020a (Mathworks, US), over the frequency range 1–65 Hz and exported to Python v3.9 where the exponent was computed. The pipeline is available here: https://github.com/fooof-tools/fooof_mat/tree/main. The aperiodic exponent was exported to R v4.2.3 for statistical analysis using the interface Rstudio v2023.03.0+386. Two subjects were excluded because of missing data.

In the animal study there was only one frontal electrode, where the 64-channel cap has multiple, so first a correlation matrix was made to test for correlations between variables for the aperiodic exponent (Fig 2). Because all the electrodes correlate significantly with each the other, they were pooled into a single exponent per individual for subsequent analysis. To further examine the correlation between age and diagnosis. Correlations were computed for each group showing a significant correlation for the schizophrenia group, but not for the healthy controls or the probable Alzheimer's group (Fig 2).

## Statistics

Correlogram was Bonferroni corrected for multiple comparisons and the color scaling was thresholded with the corrected p-value.

The statistical model for analyzing aperiodic exponents was chosen by backward elimination [19] starting with a four-way ANOVA of the model aperiodic exponent~ genotype*state*time*recording with a random effect of animal id. The recording derived variable comes from

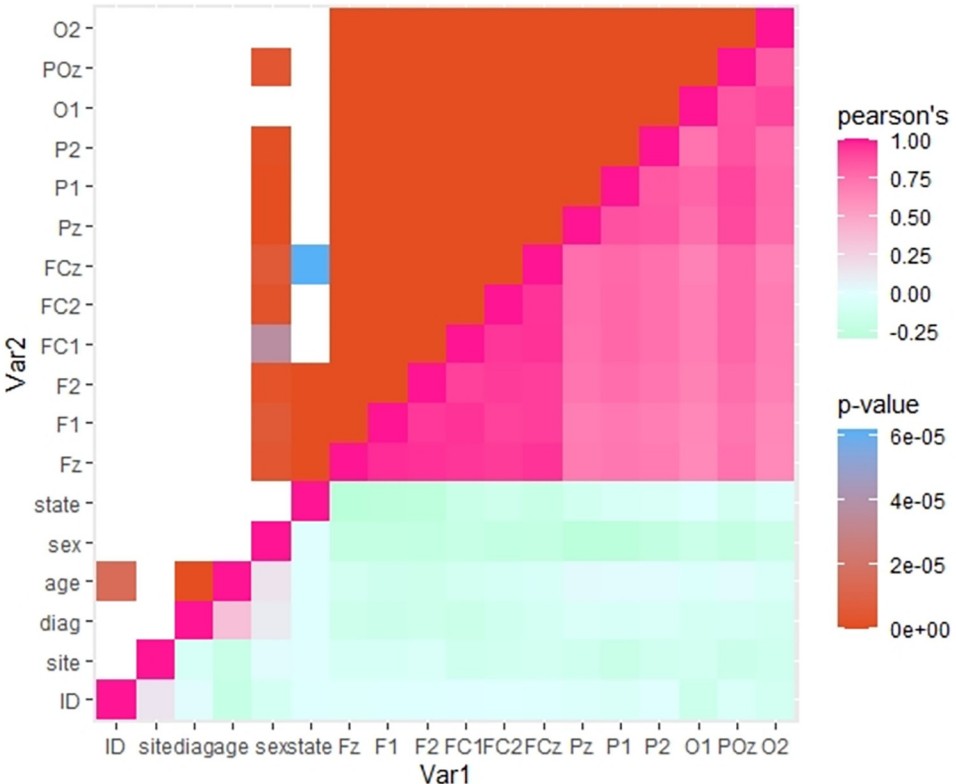

**Fig 2. Correlation matrix of relevant electrodes and variables.** The lower right: shows the correlation of variables in pink (positive) to mint (negative). The upper left: shows the Bonferroni corrected p-value (white; p≥0.05, blue; p<0.05, red; p = 0). Sex correlated with aperiodic exponent from most electrodes. State correlates significantly with Fz, F1, F2 and FCz. Age correlates significantly with ID and diagnosis.

the 24 h recordings being divided into four intervals of six hours to ease processing of the data. The four-way and three-way interactions were not statistically significant. Data is displayed as boxplots and means are written with coefficient of variance (CV). Three- or two-way ANO-VAs were carried out, to test for mean effects and followed by estimation statistics [20] for the specific effects. The outcome of estimation statistics is written as mean effect with 95 confidence intervals adjusted for multiple comparisons, in square brackets.

## Results & discussion

In the mouse data, state interacted significantly with time (F(7,1212) = 5.17, p<0.0001), recording (F(21,1212) = 3.09, p<0.0001) and genotype (F(14,1212) = 23.9, p<0.0001) (Fig 3). Since the distribution of state varies with all three factors. The lack of significant interactions between these variables is expected.

The aperiodic exponent decreased in the sequence a1>a2>a3, suggesting that the higher the frequency of definition the lower the aperiodic exponent. Additionally, the exponent decreased with genetic deficiency: wt>het>hom. Showing that the lowest aperiodic exponent emerged from the hom group in the a3 state (mean = 0.737, coefficient of variance (CV) = 0.10), while the highest is in the wt group in the a1 state (mean = 1.25, CV = 0.058). In other words, the aperiodic showed a direct relation to Nrxn1α gene expression.

To test whether aperiodic exponent also depends on vigilance state in human EEG data, the FOOOF pipeline was applied to human data from the PRISM1 project where resting state

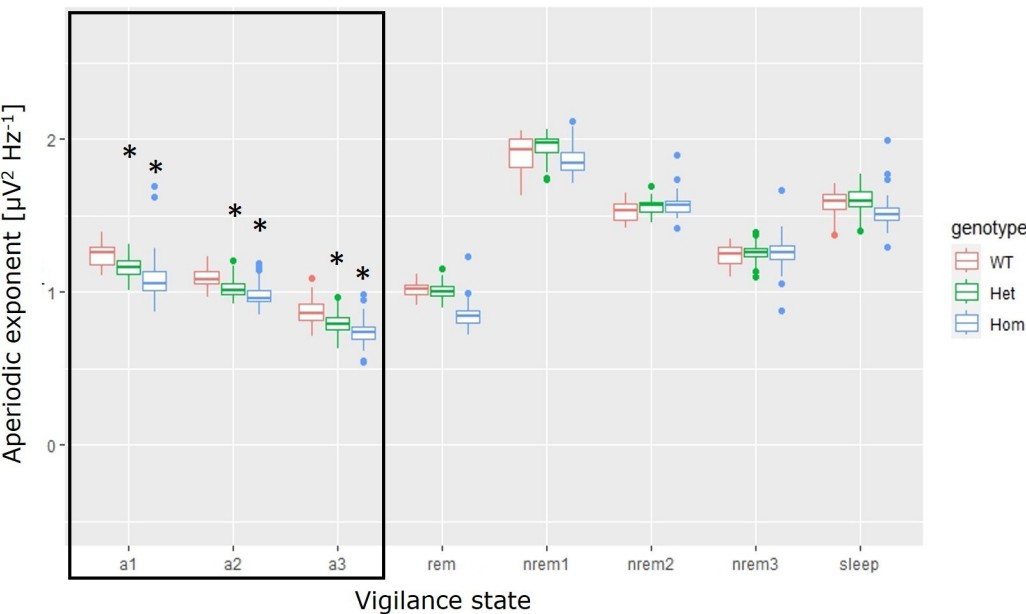

**Fig 3. Aperiodic exponent depends on state and genotype in Neurexin-1 gene knock out mice.** red: wild-type (WT), green: Heterozygous (Het), blue: Homozygous (Hom). Rabid-eye movement (rem) sleep, non-rem (nREM). The sleep category includes all nREM states in their natural order. The post-hoc only includes the active states: a1, a2 and a3. Asterisk = CI95 from estimation statistics [20] does not include zero, when compared to the WT group in the same state.

recordings were acquired from 160 individuals at five sites (two in Spain, three in the Netherlands) [14]. Apart from healthy controls (HC) the data contains individuals with probable Alzheimer's (Alz) and schizophrenia (Sz). The group sizes were: HC; n = 54, Alz; n = 52 and Sz; n = 54. For the human data the same preprocessing and analysis pipeline were used as for the preclinical data.

The PRISM1 dataset contained EEG data from 64 women and 96 men in the age range from 19–80 years. The age variable has a bimodal distribution with peaks at 25–30 and 70–75 (Fig 1). The resting state data analyzed here contains three, five-minute blocks of eyes closed, eyes open and focused on a point, and eyes closed.

Proportion of noise in the data did not interact significantly with age, sex or group. Similarly, the three awake states did not vary with demographic variables either. The average proportion of state per recording, was: noise, 5.26% (CV 0.795); a1, 61.6% CV (0.349); a2, 3.87% (2.12); a3, 29.2% (0.735). Interestingly, less than one third of the average recording was dominated by alpha band activity, though 2/3 of the recording was acquired during the eyes closed condition. This suggests that the eyes closed condition did not induce only the alpha band activity.

There was a high correlation between electrodes, therefore the electrodes are averaged into one group containing: frontal (Fz, F1, F2, FC1, FC2, FCz), parietal (Pz, P1, P2) and occipital (POz, O1, O2) electrodes (see Fig 2).

The aperiodic exponent was tested using a 3-way ANOVA of the model: aperiodic exponent ~group*state*sex, with the subject as random effect and two subjects omitted because of poor data quality. Age was not included as it is correlated with diagnosis, because no young people get the diagnosis probable Alzheimer's (mean age of Alz group = 68.6 compared to a mean age of 48.8 in the entire data set) thus age is not an independent factor. In addition, there was no

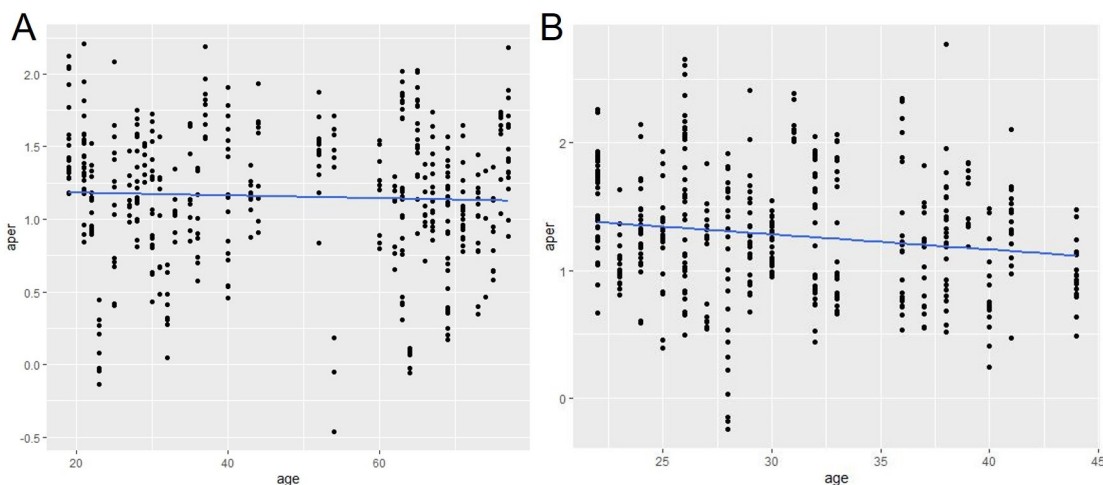

**Fig 4. Correlation of aperiodic exponent and age.** A) healthy controls B) Sz group.

correlation between aperiodic exponent and age in the healthy control group (Fig 4) therefore the control group was pooled across age.

The 5-way interaction between group, state and sex was statistically significant (Fig 5), $F_{(4,1118)} = 3.38$, p-value = 0.0093.

The post-hoc test revealed that when compared to the male healthy control group in a1 the aperiodic exponents for all female groups were significantly lower. The outcome is shown with

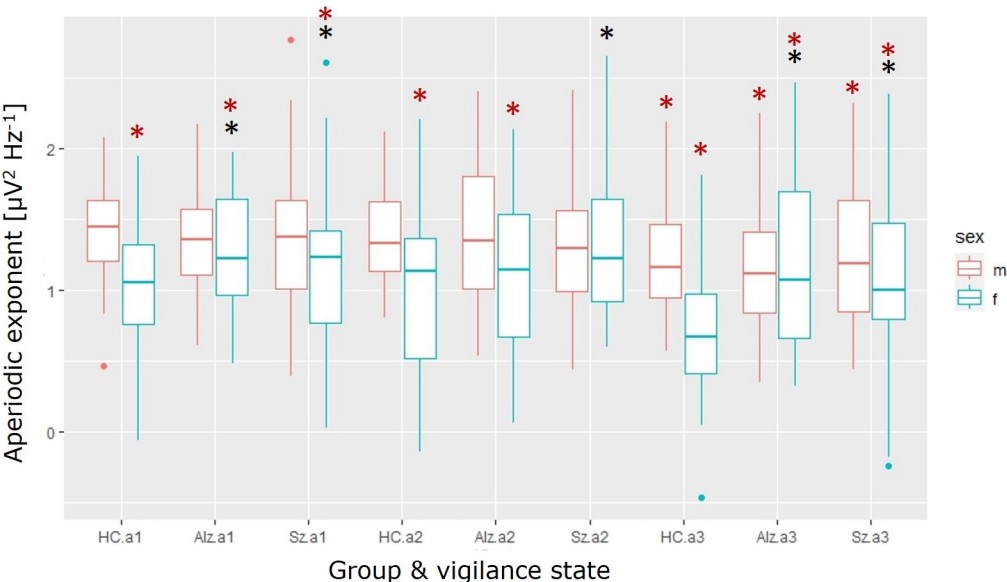

**Fig 5. Aperiodic exponent depends on state, group and sex.** red: male, blue: female. HC: healthy control, Alz: probable Alzheimer's, Sz: schizophrenia. Estimation statistics were used for the post-hoc. The red asterisk marks a statistically significant difference when compared to HC male a1. The black asterisk marks a statistically significant difference when compared to HC female in the relevant state. There is a larger variation in the human data compared to the preclinical data, which is possibly caused by a number of factors. It can be speculated that electrodes placed on the cortical surface (mice) already excludes the conductivity of skull and skin as sources of variation. Unfortunately, these sources of variation are unavoidable factors in scalp recordings (human). How much biological variation matters remains to be determined.

mean effect and 95 confidence interval in square brackets: HC (-0.411 [-0.536;-0.287]), Alz (-0.149 [-0.257;-0.0383]) and Sz (-0.206 [-0.37;-0.0334]). In the a2 state, female HC (-0.435 [-0.586;-0.286]) and Alz (-0.267 [-0.407;-0.13]) are significantly lower, while the Sz is not. In the a3 state, all groups male (HC: -0.175 [-0.272;-0.0674], Alz: -0.2 [-0.324;-0.0545], Sz: -0.171 [-0.275;-0.0625]) and female (HC: -0.701 [-0.822;-0.577], Alz: -0.221 [-0.361;-0.07], Sz: -0.307 [-0.49;-0.113]) differ from the HC male a1. When compared to the respective control group the female Sz group is higher compared to the female HC (a1: 0.205 [0.0261;0.405], a2: 0.374 [0.153;0.631], a3: 0.394 [0.191;0.607]), while the male groups did not differ. The female Alz group had a significantly higher aperiodic exponent in state a1 (0.262 [0.122;0.404]) and a3 (0.48 [0.314;0.65]). Individuals with Alzheimer's disease have previously been shown to have 'flatter' pink noise or a lower aperiodic exponent [5], which was not confirmed by the present data, possibly due to variation in samples.

In summary, the aperiodic exponent is different in both Alz and Sz women when compared to HC. Interestingly, the vigilance state alone affected the aperiodic exponent in humans in both the female and male groups with a lower exponent in a3 compared to a1. This pattern was also true for the mice. For the men, vigilance state affected the aperiodic exponent the same as the diagnosis.

To look closer at the importance of adjusting for vigilance state, it was excluded as a variable from the analysis. The two-way ANOVA of the linear model: aperiodic exponent ~diagnosis * sex, showed a significant effect of sex $F(1,154) = 12.4$, $p = 0.0006$, demonstrating that in general the female groups had a lower aperiodic exponent (Fig 6). There was no significant effect of

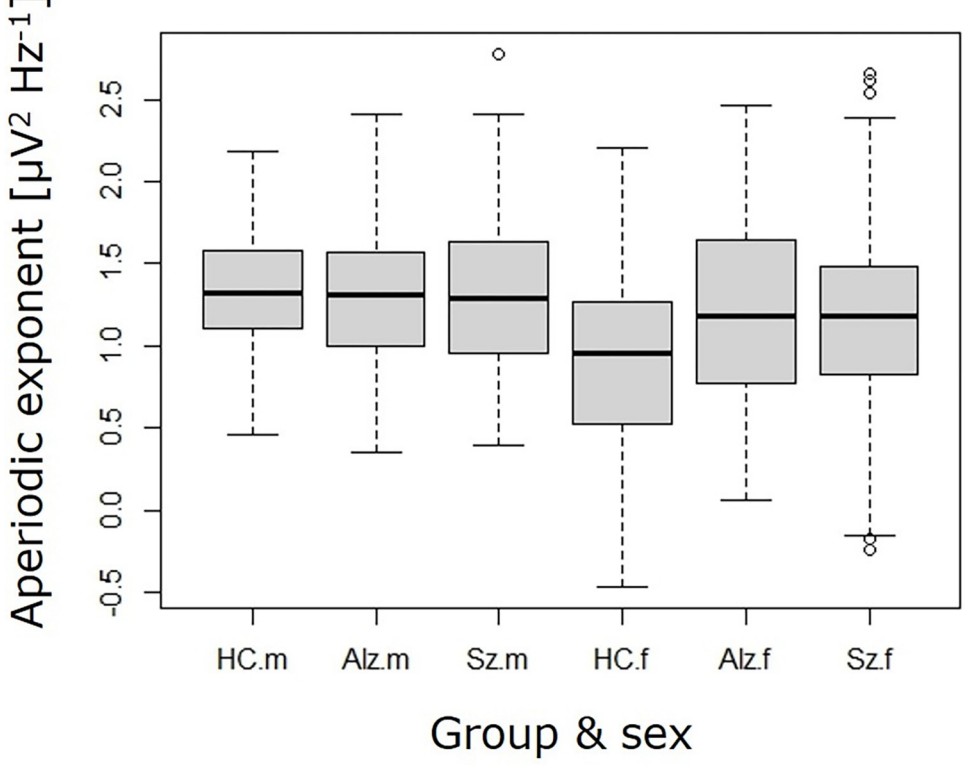

**Fig 6. No impact of diagnosis when the aperiodic exponent is not adjusted for vigilance state.** Aperiodic exponent when vigilance state is not included as a variable, showed an effect of sex and a trend for the interaction between sex and diagnosis, but no effect of diagnosis. Healthy control (HC), probable Alzheimer's (Alz), Schizophrenia (Sz), male (m), female (f).

diagnosis F(2,154) = 1.54, p = 0.218 and the interaction between diagnosis and sex F(2,154) = 2.41, p = 0.0928 although trending was not statistically significant.

Age did not interact with genotype in neither mice nor (wo)men, but in the PRISM1 dataset there was a slight but significant negative correlation between the aperiodic exponent and age in the Sz group (function: aperiodic exponent = -0.0119*age + 1.64), F(1,433) = 10.8, p-value = 0.00109, but not in the HC control group F(1,433) = 0.697, p-value = 0.404 or probable Alz group F(1,418) = 1.57, p-value = 0.211 (Fig 4).

Donoghue and colleagues show a lower exponent in older compared to younger individuals [3], while a study by van Nifterick [5], did not reveal a difference, a result similar to that found in the present study. Two other studies of the aperiodic exponent that did not control for vigilance state, have shown that the aperiodic exponent does not differ between Alz and healthy controls [21, 22], which is similar to what we find when we do not include vigilance state as a variable in the statistical test (Fig 6). The preclinical longitudinal data showed a slight, but insignificant increase in aperiodic exponent of awake states over time, which is the opposite to the findings of Donoghue and colleagues [3].

To summarize, vigilance state significantly affected the aperiodic exponent in both clinical and preclinical recordings showing a higher aperiodic exponent in the a1 state than in the a3 state, and in addition the aperiodic exponent varied with sex and diagnosis/genotype. In conclusion, vigilance state should be considered even in assessments of short durations, such as five minutes resting state eyes open/closed, where the state may be expected to be stable throughout the condition.

## Author Contributions

**Conceptualization:** Freja Gam Østergaard, Celso Arango, Inge Winter-van Rossum, Jose Luis Ayuso-Mateos, Gerard R. Dawson, Martien J. H. Kas.

**Data curation:** Brenda W. J. H. Penninx, Neetha Das.

**Formal analysis:** Freja Gam Østergaard.

**Funding acquisition:** Brenda W. J. H. Penninx, Neetha Das, Celso Arango, Gerard R. Dawson, Hugh Marston, Martien J. H. Kas.

**Investigation:** Brenda W. J. H. Penninx, Neetha Das, Celso Arango, Nic van der Wee, Inge Winter-van Rossum, Jose Luis Ayuso-Mateos, Gerard R. Dawson, Hugh Marston, Martien J. H. Kas.

**Methodology:** Freja Gam Østergaard, Nic van der Wee, Inge Winter-van Rossum, Jose Luis Ayuso-Mateos, Gerard R. Dawson, Hugh Marston.

**Project administration:** Neetha Das, Celso Arango, Martien J. H. Kas.

**Resources:** Neetha Das, Celso Arango, Nic van der Wee, Inge Winter-van Rossum, Jose Luis Ayuso-Mateos, Gerard R. Dawson, Hugh Marston, Martien J. H. Kas.

**Software:** Freja Gam Østergaard.

**Supervision:** Brenda W. J. H. Penninx, Martien J. H. Kas.

**Validation:** Freja Gam Østergaard.

**Visualization:** Freja Gam Østergaard.

**Writing – original draft:** Freja Gam Østergaard.

**Writing – review & editing:** Freja Gam Østergaard, Brenda W. J. H. Penninx, Neetha Das, Celso Arango, Nic van der Wee, Inge Winter-van Rossum, Jose Luis Ayuso-Mateos, Gerard R. Dawson, Hugh Marston, Martien J. H. Kas.

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
