## [Decision Letter · Decision Letter 0]

4 Jun 2024

PONE-D-24-10523The aperiodic exponent of neural activity varies with vigilance state in mice and menPLOS ONE

Dear Dr. Ostergaard,

Thank you for submitting your manuscript to PLOS ONE. After careful consideration, we feel that it has merit but does not fully meet PLOS ONE’s publication criteria as it currently stands. Therefore, we invite you to submit a revised version of the manuscript that addresses the points raised during the review process.

**- please try to respond and ameliorate all the concerns and issues pointed out by our reviewers**

We look forward to receiving your revised manuscript.

Kind regards,

Prof. Dr, Dragan Hrncic, MD, PhD 

Academic Editor

PLOS ONE

Journal Requirements:

This research was supported by the PRISM project (www.prism-project.eu) which has received funding from the Innovative Medicines Initiative 2 Joint Undertaking under grant agreement No 115916. This Joint Undertaking receives support from the European Union’s Horizon 2020 research and innovation program and EFPIA. This publication reflects only the authors' views neither IMI JU nor EFPIA nor the European Commission are liable for any use that may be made of the information contained therein. 

Dr. Arango has been a consultant to or has received honoraria or grants from Acadia, Angelini, Biogen, Boehringer, Gedeon Richter, Janssen Cilag, Lundbeck, Medscape, Menarini, Minerva, Otsuka, Pfizer, Roche, Sage, Servier, Shire, Schering Plough, Sumitomo Dainippon Pharma, Sunovion, Takeda and Teva.

The other authors have declared that no competing interests exist.

We note that one or more of the authors are employed by a commercial company: Acadia, Angelini, Biogen, Boehringer, Gedeon Richter, Janssen Cilag, Lundbeck, Medscape, Menarini, Minerva, Otsuka, Pfizer, Roche, Sage, Servier, Shire, Schering Plough, Sumitomo Dainippon Pharma, Sunovion, Takeda and Teva.

“The funder provided support in the form of salaries for authors, but did not have any additional role in the study design, data collection and analysis, decision to publish, or preparation of the manuscript. The specific roles of these authors are articulated in the ‘author contributions’ section.”

5. We noted in your submission details that a portion of your manuscript may have been presented or published elsewhere. "The PRISM project is a large EU funded project and the data has been used for other studies, but the results presented here are unique to this study." Please clarify whether this [conference proceeding or publication] was peer-reviewed and formally published. If this work was previously peer-reviewed and published, in the cover letter please provide the reason that this work does not constitute dual publication and should be included in the current manuscript.

Reviewers' comments:

Reviewer's Responses to Questions

**Comments to the Author**

1. Is the manuscript technically sound, and do the data support the conclusions?

Reviewer #1: Yes

Reviewer #2: No

2. Has the statistical analysis been performed appropriately and rigorously? 

Reviewer #1: Yes

Reviewer #2: I Don't Know

3. Have the authors made all data underlying the findings in their manuscript fully available?

Reviewer #1: Yes

Reviewer #2: No

4. Is the manuscript presented in an intelligible fashion and written in standard English?

Reviewer #1: Yes

Reviewer #2: Yes

5. Review Comments to the Author

**Reviewer #1:** - **Novel Contribution**:

- The paper investigates the influence of vigilance state on the 1/f signal in EEG data, addressing a gap in existing literature. It provides valuable insights into the importance of considering vigilance state during EEG analysis, which can impact the interpretation of results and the identification of biomarkers for neurological disorders.

- **Comprehensive Approach**:

- The study utilizes both animal (mouse) and human EEG data, providing a comprehensive analysis across species. This approach enhances the generalizability of the findings and allows for comparisons between preclinical and clinical settings.

- **Methodology**:

- The methodology employed in the study is robust, involving the use of the FOOOF pipeline for EEG data analysis and careful scoring of vigilance states. Additionally, ethical considerations are addressed, with animal work conducted in accordance with EU directives and human data obtained through approved protocols.

- **Significant Findings**:

- The results demonstrate significant effects of vigilance state on the 1/f signal, both in mouse models of schizophrenia and in human cohorts with schizophrenia and Alzheimer's disease. These findings underscore the importance of proper vigilance state labeling and correction in EEG analysis to accurately assess disease-related changes.

- **Implications for Clinical Practice**:

- By highlighting the impact of vigilance state on EEG signals, the paper has practical implications for clinical practice. It emphasizes the need for clinicians and researchers to carefully account for vigilance state variations, particularly in the context of neurological disorders, to ensure accurate diagnosis and treatment monitoring.

- This study delves into the impact of vigilance state on the 1/f signal within EEG data, filling a notable void in current research. By exploring this aspect, the paper offers crucial revelations regarding the necessity of factoring in vigilance state when conducting EEG analyses. Such consideration proves pivotal in ensuring the accuracy and reliability of findings, consequently refining the identification of potential biomarkers for various neurological disorders. Recognizing and incorporating this specific aspect into EEG measurements not only enhances the precision of analyses but also contributes to elevating the overall quality of EEG data interpretation and subsequent research outcomes.

- Minor Adjustments to Consider:

- It is recommended to integrate the plots within the main body of the document rather than placing them at the end.

- Additionally, the supplementary material could be included within the main document, preferably in an appendix section, for improved accessibility and coherence.

**Reviewer #2:** The authors investigated the influence of various factors (including vigilance state, sex, diagnostic group (in humans), genetics (in mice)) on the aperiodic component of resting state electroencephalography (EEG) recordings. The results showed that some of these factors were associated with differences in the aperiodic exponent, both in mice and in humans.

The manuscript is short, and I found it a confusing read. It suffers from a lack of focus, a lack of detail in the methods and left me with many questions regarding the rationale, methodology, results, and interpretation. Additionally, I believe the conclusions of the study are not strongly supported by the evidence as presented. Although the research question is interesting, I have major concerns about various aspects of the study which I have listed below:

1. Perhaps the main claim of the paper is that vigilance state influences the aperiodic exponent in both humans and mice. However, how vigilance state was defined based on the EEG data is confusing. The three awake states are labelled a1, a2, and a3 throughout, and as far as I could tell from the methods these were defined based on which frequency bands were dominant in the signal. Far more detail is needed about how the vigilance state groupings were assigned to EEG data for both humans and mice. I have not come across this way of dividing EEG data before and do not believe that this way of splitting the data would be universally accepted to index differences in vigilance state. There are many reasons why delta, theta, alpha or other frequency bands might dominate the signal at a given time, not just vigilance state.

2. In general, far more methodological details are required for the reader to fully understand how the data were analysed. For instance, please describe fully the pre-processing steps for the EEG data in both mice and humans. For the various ANOVAs performed, was multiple comparison correction applied for follow-up tests? A critical methodological detail that is missing (at least I couldn’t find it) is over which frequency range the authors fit the Fooof algorithm and which Fooof settings were selected?

3. The introduction is very short and provides no rationale for many of the analyses performed. For instance, why is data being analysed from schizophrenia and preclinical Alzheimer’s disease patients when the research question is about influence of vigilance state of the aperiodic component? Similarly, why was genotype a factor in the mouse experiment? Why is sex a variable of interest in the human experiment? The inclusion of all these (apparently irrelevant) makes the results section difficult to follow and complicates interpretation of the results. Additionally, if multi-variable ANOVA interactions are truly of interest, then it would be important to show that both experiments are sufficiently statistically powered to detect them. Is this the case?

4. Pearson’s correlations were calculated for the correlations presented in Figure S2 but some of the variables are not continuous (i.e., sex, state, site). Also, why would participant ID correlate with any of the other variables (including significantly with age)? This seems very strange.

5. Why is autism included among the key words?

6. The lack of an overall discussion makes it very difficult for the reader to gauge what the take home message(s) of the study are and how they fit into existing literature. Some highly relevant recent studies are not cited and should be. For instance, Kopcanova et al., (2024) showed that the aperiodic exponent of resting state EEG does not differentiate Alzheimer’s patients from healthy controls (see also Azami et al., 2023). Additionally, in the intro a more comprehensive coverage of relevant literature is required. For example, the following is from the intro: “It has been suggested that this may be an expression of the balance between neuronal excitation and inhibition” Who suggested this? Citation(s) required.

7. What were the criteria for excluding participants based on noisy EEG?

8. In the Figure S1 caption, what does ‘Frequency accounts for the number of electrodes’ mean?

References

Azami, H., Zrenner, C., Brooks, H., Zomorrodi, R., Blumberger, D. M., Fischer, C. E., ... & PACt-MD Study Group. (2023). Beta to theta power ratio in EEG periodic components as a potential biomarker in mild cognitive impairment and Alzheimer’s dementia. Alzheimer's Research & Therapy, 15(1), 133.

Kopčanová, M., Tait, L., Donoghue, T., Stothart, G., Smith, L., Flores-Sandoval, A. A., ... & Benwell, C. S. (2024). Resting-state EEG signatures of Alzheimer's disease are driven by periodic but not aperiodic changes. Neurobiology of Disease, 190, 106380.

6. PLOS authors have the option to publish the peer review history of their article (what does this mean?). If published, this will include your full peer review and any attached files.

Reviewer #1: **Yes: **Ioannis Iossifidis

Reviewer #2: No

---

## [Author Response · Author response to Decision Letter 0]

8 Jul 2024

Copy from 'response to reviewers'-file:

Response to reviewers:

Overall, we are very pleased with the helpful and constructive feedback from the reviewers and we have responded to each point in our rebuttal below. Our replies are written in bold font after each comment or question. 

Reviewer #1:

 - **Novel Contribution**:

- The paper investigates the influence of vigilance state on the 1/f signal in EEG data, addressing a gap in existing literature. It provides valuable insights into the importance of considering vigilance state during EEG analysis, which can impact the interpretation of results and the identification of biomarkers for neurological disorders.

- **Comprehensive Approach**:

- The study utilizes both animal (mouse) and human EEG data, providing a comprehensive analysis across species. This approach enhances the generalizability of the findings and allows for comparisons between preclinical and clinical settings.

- **Methodology**:

- The methodology employed in the study is robust, involving the use of the FOOOF pipeline for EEG data analysis and careful scoring of vigilance states. Additionally, ethical considerations are addressed, with animal work conducted in accordance with EU directives and human data obtained through approved protocols.

- **Significant Findings**:

- The results demonstrate significant effects of vigilance state on the 1/f signal, both in mouse models of schizophrenia and in human cohorts with schizophrenia and Alzheimer's disease. These findings underscore the importance of proper vigilance state labeling and correction in EEG analysis to accurately assess disease-related changes.

- **Implications for Clinical Practice**:

- By highlighting the impact of vigilance state on EEG signals, the paper has practical implications for clinical practice. It emphasizes the need for clinicians and researchers to carefully account for vigilance state variations, particularly in the context of neurological disorders, to ensure accurate diagnosis and treatment monitoring.

- This study delves into the impact of vigilance state on the 1/f signal within EEG data, filling a notable void in current research. By exploring this aspect, the paper offers crucial revelations regarding the necessity of factoring in vigilance state when conducting EEG analyses. Such consideration proves pivotal in ensuring the accuracy and reliability of findings, consequently refining the identification of potential biomarkers for various neurological disorders. Recognizing and incorporating this specific aspect into EEG measurements not only enhances the precision of analyses but also contributes to elevating the overall quality of EEG data interpretation and subsequent research outcomes.

We are grateful to the reviewer for the kind words and good reception of our work.

- Minor Adjustments to Consider:

- It is recommended to integrate the plots within the main body of the document rather than placing them at the end.

Please note that the submitted format is chosen by PlosOne for the review; In the final version of our manuscript, the figures will be placed with their captions in the typeset article.

- Additionally, the supplementary material could be included within the main document, preferably in an appendix section, for improved accessibility and coherence.

We thank the reviewer for the good idea, the supplementary figures have now been incorporated in the methods and results section.

Reviewer #2: The authors investigated the influence of various factors (including vigilance state, sex, diagnostic group (in humans), genetics (in mice)) on the aperiodic component of resting state electroencephalography (EEG) recordings. The results showed that some of these factors were associated with differences in the aperiodic exponent, both in mice and in humans.

The manuscript is short, and I found it a confusing read. It suffers from a lack of focus, a lack of detail in the methods and left me with many questions regarding the rationale, methodology, results, and interpretation. Additionally, I believe the conclusions of the study are not strongly supported by the evidence as presented. Although the research question is interesting, I have major concerns about various aspects of the study which I have listed below:

1. Perhaps the main claim of the paper is that vigilance state influences the aperiodic exponent in both humans and mice. However, how vigilance state was defined based on the EEG data is confusing. The three awake states are labelled a1, a2, and a3 throughout, and as far as I could tell from the methods these were defined based on which frequency bands were dominant in the signal. Far more detail is needed about how the vigilance state groupings were assigned to EEG data for both humans and mice. I have not come across this way of dividing EEG data before and do not believe that this way of splitting the data would be universally accepted to index differences in vigilance state. There are many reasons why delta, theta, alpha or other frequency bands might dominate the signal at a given time, not just vigilance state.

The definition of vigilance states is based on previous works, as written in the methods section p. 6 l. 134-136. ‘The mouse data was scored into seven vigilance states (three active states, REM and three nREM states), and the sleep category containing the nREM cycles in their natural sequence (figure 3). The active states are dominated by specific frequency bands. A1 is dominated by delta band activity(15) (0.2-3 Hz), a2 by theta band activity(16,17) (3.2-6 Hz) and a3 by alpha band activity (6.2-12 Hz). A3 is more commonly known as resting state, while a1 is awake and active, and a2 has been associated with hippocampal activity(17).’ References can be found in the manuscript.

The area has in general received very little attention. A point made in the paper is that the awake vigilance states are poorly described in human data, as the focus of most awake recordings have been on various stimulus-driven effects.

With regards to the following statement ‘There are many reasons why delta, theta, alpha or other frequency bands might dominate the signal at a given time, not just vigilance state.’ That would be true on a single cell level, but when we’re talking local field potentials then the frequency band recorded over the frontal cortex indicates a background for all other processing, which would entail a vigilance state.

2. In general, far more methodological details are required for the reader to fully understand how the data were analysed. For instance, please describe fully the pre-processing steps for the EEG data in both mice and humans. 

A comprehensive description of the data analysis pipeline has now been added to the method section of the revised version of our manuscript. L. 129-144. ‘Data pipeline

All data was filtered using a butterworth filter of order 4 and high-pass: 0.5 Hz and low-pass: 70 Hz. Artifacts were rejected if they deviated more than 3 SD from the mean of the signal. The signal was scored into vigilance states on the basis of 1 s epochs. The mouse data was scored into seven vigilance states (three active states, REM and three nREM states), and the sleep category containing the nREM cycles in their natural sequence (figure 31). The active states are dominated by specific frequency bands. A1 is dominated by delta band activity(15) (0.2-3 Hz), a2 by theta band activity(16,17) (3.2-6 Hz) and a3 by alpha band activity (6.2-12 Hz). A3 is more commonly known as resting state, while a1 is awake and active, and a2 has been associated with hippocampal activity(17). The human data only included awake states and was scored into a1, a2 and a3.

After scoring the data the aperiodic exponent was computed using FOOOF described by Donoghue and colleagues. Power spectral density was computed in Matlab v2020a (Mathworks, US) and exported to Python v3.9 where the exponent was computed. The pipeline is available here: https://github.com/fooof-tools/fooof_mat/tree/main. The aperiodic exponent was exported to R v4.2.3 for statistical analysis using the interface Rstudio v2023.03.0+386. Two subjects were excluded because of missing data.’

For the various ANOVAs performed, was multiple comparison correction applied for follow-up tests? 

The confidence intervals of the estimation statistics used for post-hoc were adjusted for multiple comparison. L. 172-173 ‘The outcome of estimation statistics is written as mean effect with 95 confidence intervals adjusted for multiple comparisons, in square brackets.’

A critical methodological detail that is missing (at least I couldn’t find it) is over which frequency range the authors fit the Fooof algorithm and which Fooof settings were selected?

The aperiodic component was fitted for each state (instead of channel) over PSD in the range 1-65 Hz, and exported from matlab to Python, as described in the Github tutorial for the matlab wrapper: https://github.com/fooof-tools/fooof_mat/blob/main/examples/fooof_example_multi_spectra.m

No additional parameters/settings were chosen.

3. The introduction is very short and provides no rationale for many of the analyses performed. For instance, why is data being analysed from schizophrenia and preclinical Alzheimer’s disease patients when the research question is about influence of vigilance state of the aperiodic component? 

A key point made in the article is that, not adjusting for vigilance state can mask biomarkers for psychiatric conditions, this has been elaborated in the introduction, demonstrated in the figure 6 and repeated in the last part of result and discussion section.

Similarly, why was genotype a factor in the mouse experiment? 

A sentence has been added to the introduction: mentioning that genotype in mice is used as a surrogate for diagnosis in the animals. L. 53-54 ‘As animals cannot be diagnosed per se, genotype serves as a surrogate for diagnosis.’

Why is sex a variable of interest in the human experiment? 

It is well-known that psychiatric symptoms vary between sexes (see ref. 7 of the manuscript), these differences may stem from underlying mechanistic differences. Therefore, we decided to test the significance of the variable. 

The inclusion of all these (apparently irrelevant) makes the results section difficult to follow and complicates interpretation of the results. Additionally, if multi-variable ANOVA interactions are truly of interest, then it would be important to show that both experiments are sufficiently statistically powered to detect them. Is this the case?

The multi-variate ANOVA interactions are not of specific interest. Testing the highest-level interaction is the first step in backward elimination described by Hocking (ref 18), which is a data-driven approach to ensure the validity of the statistical model chosen. This should have been stated in the methods and has therefore been added. L.166-168 ‘The statistical model for analyzing aperiodic exponents was chosen by backward elimination (18) starting with a four-way ANOVA of the model aperiodic exponent~ genotype*state*time*recording with a random effect of animal id. The…’

4. Pearson’s correlations were calculated for the correlations presented in Figure S2 but some of the variables are not continuous (i.e., sex, state, site). 

The correlogram is used to point out general patterns in the data (including the categorical variables). 

Also, why would participant ID correlate with any of the other variables (including significantly with age)? This seems very strange.

The participant ID is per site and it shows that older participants were recruited first. It is irrelevant for the study, but still a pattern in the data. To be transparent about the data, then it is left in there.

5. Why is autism included among the key words?

The NRXN1 is found in GWAS studies of both schizophrenia and autism, this has been elaborated in the introduction. L.45-55. ‘The mouse model used in the present study has a knockout of Nrxn1α. Loss-of-function mutations of the NRXN1 gene are related to autism as well as schizophrenia(11).’

6. The lack of an overall discussion makes it very difficult for the reader to gauge what the take home message(s) of the study are and how they fit into existing literature. Some highly relevant recent studies are not cited and should be. For instance, Kopcanova et al., (2024) showed that the aperiodic exponent of resting state EEG does not differentiate Alzheimer’s patients from healthy controls (see also Azami et al., 2023). 

We thank the reviewer for the references; they are very relevant in highlighting that potential biomarkers may be overlooked when data is not adjusted for vigilance state. The suggested references have now been included in the section that combines results and discussion.

Additionally, in the intro a more comprehensive coverage of relevant literature is required. For example, the following is from the intro: “It has been suggested that this may be an expression of the balance between neuronal excitation and inhibition” Who suggested this? Citation(s) required.

It is stated in a 2023 paper by van Nifterick and the 2017 paper by Gao. Both have now been emphasized in the introduction of the revised manuscript. L.41-42. ‘This slope is also referred to as the aperiodic exponent. It has been suggested that this may be an expression of the balance between neuronal excitation and inhibition(5,6).’

7. What were the criteria for excluding participants based on noisy EEG?

When the FOOOF pipeline gave a Nan (Not a number), the data was dropped from the correlogram.

8. In the Figure S1 caption, what does ‘Frequency accounts for the number of electrodes’ mean?

The statement is referring to the y-axis. This has now been elaborated in the caption in the revised version of the manuscript. ‘Age distribution in the PRISM1 data set. Age is shown in years, but binned per five years. Frequency (y-axis) accounts for the number of electrodes (12 per participant).’

---

## [Decision Letter · Decision Letter 1]

26 Jul 2024

The aperiodic exponent of neural activity varies with vigilance state in mice and men

PONE-D-24-10523R1

Dear Dr. Ostergaard,

We’re pleased to inform you that your manuscript has been judged scientifically suitable for publication and will be formally accepted for publication once it meets all outstanding technical requirements.

Kind regards,

Prof. Dr. Dragan Hrncic, MD, PhD

Academic Editor

PLOS ONE

Additional Editor Comments (optional):

Reviewers' comments:

Reviewer's Responses to Questions

**Comments to the Author**

1. If the authors have adequately addressed your comments raised in a previous round of review and you feel that this manuscript is now acceptable for publication, you may indicate that here to bypass the “Comments to the Author” section, enter your conflict of interest statement in the “Confidential to Editor” section, and submit your "Accept" recommendation.

Reviewer #1: All comments have been addressed

Reviewer #2: All comments have been addressed

2. Is the manuscript technically sound, and do the data support the conclusions?

Reviewer #1: Yes

Reviewer #2: Yes

3. Has the statistical analysis been performed appropriately and rigorously? 

Reviewer #1: Yes

Reviewer #2: Yes

4. Have the authors made all data underlying the findings in their manuscript fully available?

Reviewer #1: Yes

Reviewer #2: Yes

5. Is the manuscript presented in an intelligible fashion and written in standard English?

Reviewer #1: Yes

Reviewer #2: Yes

6. Review Comments to the Author

Reviewer #1: (No Response)

Reviewer #2: Thank you for addressing my initial comments and concerns. I have no further comments to make on the manuscript.

7. PLOS authors have the option to publish the peer review history of their article (what does this mean?). If published, this will include your full peer review and any attached files.

Reviewer #1: **Yes: **Ioannis Iossifidis

Reviewer #2: No

---

## [Editor Report · Acceptance letter]

1 Aug 2024

PONE-D-24-10523R1 

PLOS ONE

Dear Dr. Ostergaard, 

I'm pleased to inform you that your manuscript has been deemed suitable for publication in PLOS ONE. Congratulations! Your manuscript is now being handed over to our production team.

Kind regards, 

on behalf of

Professor Dragan Hrncic 

Academic Editor

PLOS ONE